# BERTs are Generative In-Context Learners

**David Samuel**
Language Technology Group
University of Oslo
davisamu@uio.no

## Abstract

While in-context learning is commonly associated with causal language models, such as GPT, we demonstrate that this capability also 'emerges' in masked language models. Through an *embarrassingly simple* inference technique, we enable an existing masked model, DeBERTa, to perform generative tasks without additional training or architectural changes. Our evaluation reveals that the masked and causal language models behave very differently, as they clearly outperform each other on different categories of tasks. These complementary strengths suggest that the field's focus on causal models for in-context learning may be limiting – both architectures can develop these capabilities, but with distinct advantages; pointing toward promising hybrid approaches that combine the strengths of both objectives.

## 1   Introduction

Masked language models used to dominate the field of natural language processing due to their adaptability across diverse tasks and their superior performance compared to causal language models (Radford et al., 2018; Devlin et al., 2019). Between 2018 and 2020, the field witnessed a surge in the development of these models (Devlin et al., 2019; Liu et al., 2019; Lan et al., 2020, inter alia). However, the field dramatically shifted with GPT-3 and its introduction of *in-context learning* – the ability to infer and perform tasks from prompts and examples without any finetuning (Brown et al., 2020). This capability eliminated the need for task-specific training data and deep-learning expertise, making such models far more practical for real-world applications. This perceived advantage led many researchers and practitioners to abandon masked language models in favor of GPT-style architectures.

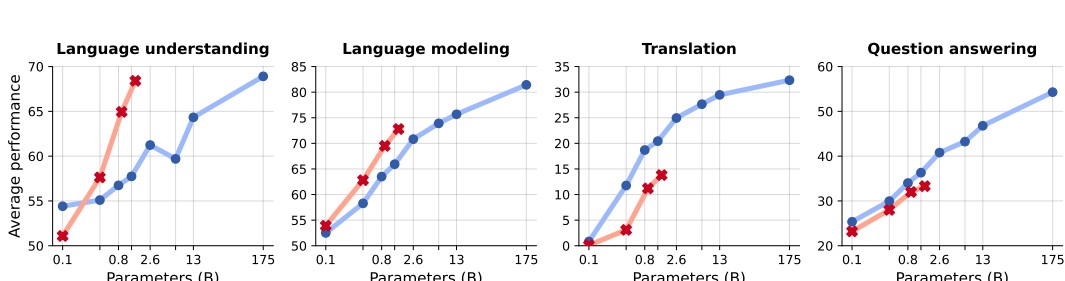

Figure 1: **The average 1-shot performance across four groups of NLP tasks**   We compare the scaling abilities of DeBERTa (four sizes in red) with GPT-3 (eight sizes in blue). Even though these models rely on different training objectives, they scale in a similar log-linear manner overall. Yet, on a task-by-task basis, the pretraining methods lead to substantial differences between them.

38th Conference on Neural Information Processing Systems (NeurIPS 2024).

Previous studies of 'emergent' in-context learning abilities have focused almost exclusively on causal language models, creating a widespread assumption that this capability is unique to them (Saunshi et al., 2021; Olsson et al., 2022; Wei et al., 2022; Wang et al., 2023, inter alia). In this paper, we challenge this assumption by demonstrating that in-context learning can emerge in masked language models as well. In-context learning is a more general phenomenon and should not be studied with a singular pretraining objective in mind. Moreover, the assumed inability of masked language models to perform (generative) in-context learning has rendered them outdated – as explicitly noted by Tay et al. (2023): *"BERT-style models are very restricted in their generative capabilities. Because of the cumbersomeness of task-specific classification heads, we strongly do not recommend using this class of autoencoding models moving forward and consider them somewhat deprecated."*

In this paper, we challenge these prevailing assumptions about masked language models (MLMs). We present empirical evidence showing that DeBERTa, an MLM released just one month after GPT-3, is equally adept at in-context learning. Our findings suggest that the capacity for in-context learning is not tied to the training objective, but can be achieved across different types of language models. To our surprise, we found that DeBERTa does not simply mimic the performance of GPT-3 – the two model behave very differently – DeBERTa is clearly much better on tasks such as language understanding, and, on the other hand, much worse on tasks such as closed-book question answering. This suggests that masked and causal language modeling are two complementary training objectives and that there is a great potential for a training method that combines the strengths of both objectives. Finally, *scaling* (performance improvement with increased size of pretrained language models) is a crucial feature of modern language models; we demonstrate that MLMs do *scale* on in-context learning (Figure 1).

We introduce a simple inference technique that transforms an MLM into a generative model without any further training. Using publicly available DeBERTa checkpoints, we show that the MLM training objective not only provides a versatile way of encoding text, but is also competitive in text generation and text completion ranking. This claim is tested by following the same evaluation suite as GPT-3, speculating on an 'alternative reality' in which a masked language model is the first model reported to achieve the so-called 'emergent' in-context learning abilities. While other masked language models could potentially demonstrate similar capabilities, we deliberately target DeBERTa because of its large size and its length-generalization abilities. Ultimately, our goal is to demonstrate that MLMs *can* perform in-context learning and that they *can* be surprisingly good at doing so.

**Outline**  First, Section 2 (Method) describes the inference methods used to evaluate the in-context learning abilities of an off-the-shelf masked language model. Then Section 3 (DeBERTa) describes the details of the particular model used in this study. Section 4 (Evaluation) details the evaluation setup and compares DeBERTa with GPT-3. Finally, Section 5 (Related work) talks about other relevant work within this topic, and the paper concludes with Section 6 (Conclusion).

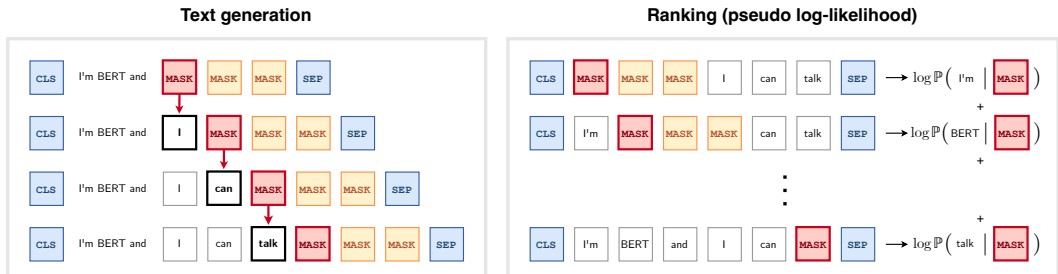

Figure 2: **Illustration of the proposed methods for using a masked language model for text generation and text ranking**  We show how to adapt a masked language model for in-context-learning tasks through simple input reformatting, requiring no additional training. LEFT: Text generation is achieved by 1) appending [MASK] tokens to the input prompt, 2) predicting the next token for the first mask, and 3) iteratively appending new masks and predicting tokens. RIGHT: A similar approach is used to retrieve a pseudo-log-likelihood score of a text sequence that can be used to rank multiple sequences by their individual likelihoods. Both methods maintain the model's original architecture while enabling new capabilities through careful input formatting.

## 2 Method: text generation and ranking with masked language models

The goal of this article is to reuse an existing pretrained masked language model for (generative) in-context learning. We achieve this without any additional training or finetuning, our method only slightly changes the sequence of input tokens, as illustrated in Figure 2. There are two methods used to solve tasks with in-context learning: **text generation** where the model completes a given prompt (e.g. for translation) and **ranking** where the model chooses an answer from several options (e.g. for multiple choice questions).

### 2.1 Text generation

Masked language models are trained on semi-supervised fill-in-the-blanks tasks and so they cannot be used to generate straight out of the box. One possibility is to interpret these models as Markov random fields and produce text by Gibbs sampling (Wang and Cho, 2019). However, a simpler and more consistent way to produce text is to do the familiar left-to-right autoregressive generation – we could place a [MASK] token next to a text prompt and let the model generate next token by unmasking the appended token – then, when we repeat this process in a loop, we can generate text in the same way as causal language models (and apply the same advanced generation techniques).

This straightforward inference scheme would be enough if the pretraining process were designed with this use case in mind. However, since our goal is to repurpose an existing masked language model, we have to complicate the method with two modifications that are also illustrated in Figure 2:

1. Masked language models are typically trained with a special end-of-sequence [SEP] token. This token is always present during pretraining and so we also have to include it as the last token during inference.

2. However, the addition of this end-of-sequence token creates a problem – it raises the probability that the masked token should end the sequence (for example with a full stop). Thus, in order to obtain a less restricted continuation, we include additional [MASK] tokens to pad the space in front of the end-of-sequence token. Specifically, we use two additional masks for the DeBERTa models.[1] This decision is later ablated in Appendix B.

In the end, this approach gives a probability distribution over the next token prediction, thus we can use any existing method for searching or sampling an output sequence. We follow GPT-3 and use beam search with four candidate beams for all generative tasks.

**Limitations**   While this method works with the same quadratic time complexity (in sequence length), it is slower in practice because it is not possible to cache the intermediate self-attention key and value vectors. Instead, these have to be recomputed every step due to the bidirectional nature of the model. While our current implementation prioritizes demonstrating the core capability over optimization, several promising approaches could address these computational limitations in future work. For example, using prefix language modeling or selectively updating hidden vectors could significantly improve efficiency. We leave these optimizations for future work to maintain focus on establishing the fundamental ability of MLMs to generate text.

### 2.2 Ranking

Many of the existing tasks for evaluating LLMs can be formulated as classification tasks where models have to select the correct answer from a number of different options. Brown et al. (2020) rank the candidate completions based on their estimated conditional log-likelihood, which can be computed exactly by the chain rule (where $w_0 \oplus w_1 \ldots w_k$ is a completion of a prompt $c$):

$$\log \mathbb{P}(w_0 \oplus w_1 \ldots w_k \,|\, c) = \sum_{i=0}^{k} \log \mathbb{P}(w_i \,|\, c \oplus w_0 \ldots w_{i-1}) \tag{1}$$

While this equation matches the training objective of causal language models, it is not suitable for masked language models because they are not trained to estimate $\mathbb{P}(w_i \,|\, c \oplus w_0 \ldots w_{i-1})$. Instead,

---

[1]Note that this is not an arbitrary number but it is model-specific – DeBERTa models were pretrained to unmask *spans* of masked tokens where the longest allowed spans are three tokens long (He et al., 2021).

Wang and Cho (2019) proposed to modify Equation (1) to make it more appropriate for BERT-like models. Salazar et al. (2020) then empirically showed that the resulting pseudo-log-likelihood (PLL) score can be used to accurately rank text sequences by their likelihood. Specifically, the PLL score is approximately proportional to the conditional probability of a text sequence and is computed as:

$$\log \mathbb{P}(w_0 \oplus w_1 \ldots w_k \,|\, c) \underset{\sim}{\propto} \sum_{i=0}^{k} \log \mathbb{P}(w_i \,|\, c \oplus w_0 \ldots w_{i-1} \oplus \texttt{[MASK]} \oplus w_{i+1} \ldots w_k) \qquad (2)$$

However, this approximation gets very inaccurate when there are strong local dependencies between tokens. As a counterexample, the estimated likelihood of the multi-token word '*supercalifragilistic-expialidocious*' is seven orders of magnitude higher than that of the single-token word '*super*', which is clearly an incorrect estimation of the relative frequencies of these words.[2]

We improve on this behavior by interpolating between the mathematically correct unidirectional derivation in Equation (1) and the bidirectional approximation in Equation (2). Our approach is to simply mask two additional tokens in the right context to reduce the effect of local dependencies while still taking into account the global bidirectional context. This process is illustrated in Figure 2. We conduct an ablation study of this approach in Appendix C.

**Limitations**  Even though Equations (1) and (2) look similar, the later sum is substantially more compute intensive when calculated with a transformer architecture – for a token sequence of length $k$, the first equation can be computed with passing a single sequence through a language model, while the second equation needs $k$ sequences to be processed. However, Salazar et al. (2020) showed that the PLL score can be accurately estimated in a single pass after a short self-supervised finetuning.

### 2.3 Length generalization

A potentially limiting factor of using BERT-like models is that they are typically pretrained on shorter sequences than causal language models – arguably because the training of modern causal models is already optimized for in-context learning, which requires processing of long few-shot prompts. DeBERTa is not an exception to such pretraining; it was only trained with a relatively short maximum sequence length of 512 tokens (He et al., 2021). Fortunately, the architecture of DeBERTa can easily process much longer sequences than seen during training due to its use of relative positional embeddings with logarithmic buckets (Raffel et al., 2020).

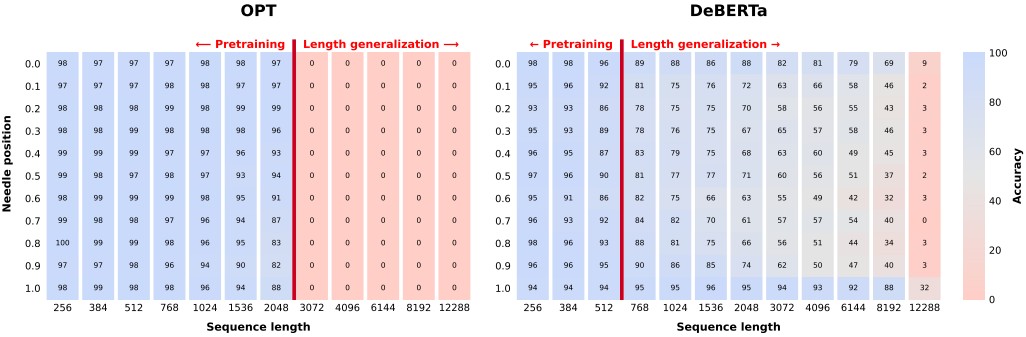

Figure 3: **Length generalization measured with a 'needle in a haystack' benchmark**  The $x$-axis indicates the total size of the 'haystack' and the $y$-axis indicates the position of the 'needle'; the values show the average exact-match accuracy for a particular configuration. Unfortunately, GPT-3 is a closed-source model and the original version is not accessible, so we use an open-source replication of GPT-3, OPT by Zhang et al. (2022), which should perform similarly on this task because of the the same transformer architecture as GPT-3. In particular, it uses absolute positional encoding, which strictly limits any model from generalizing to longer inputs than trained on.

---

[2]This is because the tokenizer splits the long word into 9 subwords and each of them is assigned an almost certain likelihood given the bidirectional context. The largest 1.5B DeBERTa estimates the pseudo log-likelihood of the first word to $-2.1$ while the second (very common) word has pseudo log-likelihood of $-9.6$.

We measure the extent to which DeBERTa generalizes to longer sequences with the 'needle in a haystack' test from RULER (Hsieh et al., 2024). Specifically, in our formulation of this task, a random 6-digit number (needle) is hidden in a long collection of essays (haystack). We then measure the exact-match accuracy of retrieving the hidden number given two variables: the total sequence length and the position of the needle in the haystack (more details about the evaluation setup are given in Appendix E.1).

The results in Figure 3 demonstrate that DeBERTa generalizes to sequences well beyond its training length, which is enabled by its relative positional encoding. For comparison, we also show results from OPT (Zhang et al., 2022), which uses absolute positional encoding like GPT-3. As expected from models using absolute positional encoding, performance drops sharply beyond the training length. This comparison highlights the importance of positional encoding choice for length generalization, independent of whether the model is masked or causal. In practice, this observation means that DeBERTa should be able to handle as many task demonstrations as models trained with longer sequences.

## 3 DeBERTa family of language models

This study uses the largest openly available English masked language model, DeBERTa with 1.5 billion parametrs, and its smaller configurations – 0.1B, 0.4B and 0.9B (He et al., 2021). DeBERTa is an improved version of a BERT language model (Devlin et al., 2019) that uses an advanced attention mechanism with relative positional embeddings – apart from being trained on a larger corpus and with a larger number of training steps.

**Training corpus**    Compared to GPT-3 and modern large language models, DeBERTa was pretrained on a relatively small and clean text corpus – totalling 78GB of data after deduplication, the corpus is comprised of the English Wikipedia (12GB), BookCorpus (6GB; Zhu et al., 2015), OpenWebText (38GB; Gokaslan and Cohen, 2019), and STORIES (31GB; Trinh and Le, 2019). This is almost an order of magnitude less data than what was used to pretrain GPT-3. Notably, our strong results – despite this data disparity – could suggest that masked language models are more data-efficient than causal models for developing in-context learning capabilities. This claim would however need to be evaluated with a comprehensive study. In comparison, GPT-3 uses 570GB of filtered CommonCrawl, WebText2 (roughly 26GB), two web-scraped book corpora (roughly 17GB and 76GB), and the English Wikipedia (roughly 4GB, estimated from Brown et al. (2020)).

**Total training compute**    Interestingly, even though DeBERTa uses a substantially smaller training corpus, it is trained on more than three times more tokens than GPT-3 (1 trillion compared to 300 billion).[3] However, the loss is computed only on 15% of tokens (150 billion) and it is not clear what would be the effective number of tokens used for pretraining. Nevertheless, the total compute used for training depends on the number of input tokens and it is roughly $8.0 \cdot 10^{21}$ FLOPs for the 1.5B DeBERTa, and $2.4 \cdot 10^{21}$ FLOPs for the 1.3B GPT-3.

**Causal conversion for HuggingFace**    We have converted the officially available DeBERTa checkpoint into a HuggingFace (Wolf et al., 2020) implementation of `AutoModelForCausalLM` (following the method in Section 2.1), and released it openly at `https://hf.co/ltg/deberta-xxlarge-fixed`. The weights of this model are exactly the same as the official release from `microsoft/deberta-v2-xxlarge`, but we have fixed some bugs found in the original modeling script in addition to implementing the text generation abilities.[4] Similarly, we have also converted the smaller DeBERTa models and released them as `ltg/deberta-base-fixed`, `ltg/deberta-large-fixed`, and `ltg/deberta-xlarge-fixed`.

---

[3]This means that DeBERTa was trained on dozens of repetitions of its training corpus, not unlike other popular masked language models (Devlin et al., 2019; Liu et al., 2019) – suggesting that this type of models operates under different 'training laws' than causal language models (Muennighoff et al., 2023).

[4]Specifically: 1) incorrect name of the output embedding weights in the checkpoint file, 2) non-functioning implementation of the enhanced mask decoder (EMD), and 3) missing truncation of the relative positional indices.

# 4 Evaluation

As our goal is to compare two language models released around the same time in 2020 – GPT-3 and DeBERTa – we replicate the evaluation setup used for GPT-3 (Brown et al., 2020) and apply it to the latter model. This also means that we follow GPT-3 and divide the tasks into generative ones (such as machine translation) and into classification tasks (such as BoolQ) – the first group uses the method described in Section 2.1 and the second type of task uses the ranking described in Section 2.2. Generation is performed with beam search (4 candidate beams), and ranking uses the modified PLL scores (and the normalized unconditional probability of completions $\frac{\mathbb{P}(\text{completion} \mid \text{context})}{\mathbb{P}(\text{completion} \mid \text{answer context})}$ for ARC and OpenBookQA), again replicating the choices for GPT-3). We also use the exact same prompt templates, with the exception of the machine translation task – its template did not produce any meaningful output, and so we decided to use the simple prompt template from Garcia et al. (2023) instead. More details on the evaluation setup can be found in Appendix E. Note that using prompts optimized for GPT-3 is slightly unfair to all other models, as prompting has a strong influence on performance (Gonen et al., 2023), but we believe that it makes the results more convincing than if we were to do extensive prompt engineering.

To show the strengths and weaknesses of DeBERTa in (generative) in-context learning, we evaluate it on four groups of tasks and compare it to the results from Brown et al. (2020). The four groups are language understanding (SuperGLUE), language modeling (text completion and Winograd-like tasks), machine translation, and question answering (closed-book question answering and commonsense reasoning). We detail each of these groups of tasks below.

Before looking into the details of each group, we show the overall aggregated scores for each group in Figure 1 and Figure 4. The first figure shows how the performance of both models scales with their size, while the latter figure compares the in-context learning abilities of the two language models. We also provide a qualitative evaluation of text generation in Appendix A and full results in Appendix F.

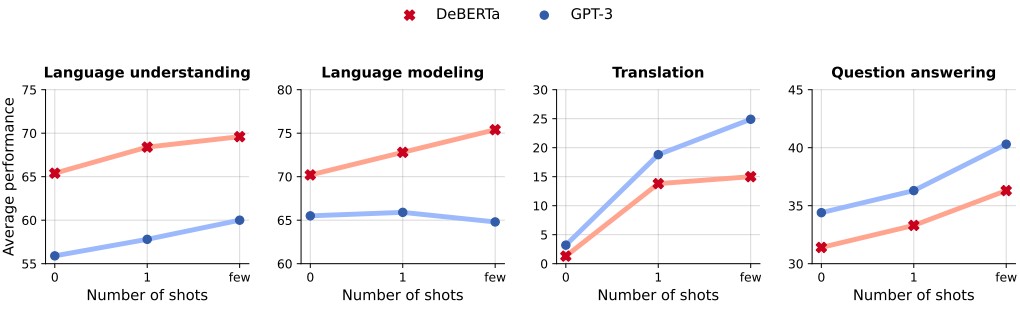

Figure 4: **The performance improvement with increased number of in-context examples**   We compare the in-context learning ability of 1.5B DeBERTa (in red) with 1.3B GPT-3 (in blue) using prompts without any completed examples (0-shot), prompts with a single randomly sampled gold sample (1-shot), and prompts with *few* examples (4 – 64 examples, depending on the task). This figure demonstrates that a masked language model behaves similarly to a causal language model in the in-context learning regime. More detailed few-shot evaluation is in Figure 5.

## 4.1 Language understanding (SuperGLUE)

We use SuperGLUE (Wang et al., 2019) as a popular collection of standard NLP tasks, allowing us to evaluate the performance on different aspects of natural language understanding.

In total, this benchmark consists of eight datasets, selected to be difficult for the contemporary (finetuned) language models. The Boolean Questions dataset is a yes/no reading comprehension dataset evaluated with accuracy (BoolQ; Clark et al., 2019); CommitmentBank is a three-class textual entailment dataset evaluated with accuracy and $F_1$ score, where the multi-class $F_1$ is computed as the unweighted average of the $F_1$ per class (CB; de Marneffe et al., 2019); the Choice of Plausible Alternatives dataset is a causal reasoning task evaluated with accuracy (COPA; Roemmele et al.,

2011); Multi-Sentence Reading Comprehension is a multiple choice dataset, evaluated with exact-match (of all answers per question) accuracy and $F_{1\alpha}$ score computed over all flattened answers (MultiRC; Khashabi et al., 2018); Reading Comprehension with Commonsense Reasoning Dataset is another reading comprehension dataset, it is evaluated with the exact-match accuracy and token-level $F_1$ score (ReCoRD; Zhang et al., 2018); the collection of Recognizing Textual Entailment datasets is a textual entailment task evaluated with accuracy (RTE; Dagan et al., 2006; Bar-Haim et al., 2006; Giampiccolo et al., 2007); the Word-in-Context dataset is a word sense disambiguation dataset evaluated with accuracy (WiC; Pilehvar and Camacho-Collados, 2019); and finally, the Winograd Schema Challenge evaluates coreference resolution capabilities (WSC; Levesque et al., 2012).

**Results** We show the resulting scores of evaluation with the same prompts as GPT-3 in Table 1 and Appendix D. DeBERTa clearly outperforms its contemporary and scales much more favorably than the family of GPT models (Figure 1). Interestingly, the average performance of the 1.5B DeBERTa gets close to the reported performance of the largest 175B GPT-3 (68.4 vs. 68.9, 1-shot). However, this average score is still far from the performance of a *finetuned* DeBERTa, which is more than 20 percentage points higher (He et al., 2021); the average few-shot performance of DeBERTa is slightly better than a finetuned BERT-large (Devlin et al., 2019; Wang et al., 2019).

Table 1: **Natural language understanding results** All results in this table are evaluated with accuracy (higher is better). The table shows the performance of the largest available DeBERTa (1.4 billion parameters) and of a similarly-sized GPT-3 model, the best results are boldfaced. The average score is calculated over averaged task scores (in case a task uses more than one metric).

|  | BoolQ | CB | COPA | MultiRC | ReCoRD | RTE | WiC | WSC | Average |
|---|---|---|---|---|---|---|---|---|---|
| *0-shot* | | | | | | | | | |
| GPT-3 | 62.4 | 19.6 | 77.0 | **13.6** | 84.1 | 56.0 | 50.0 | 61.5 | 55.9 |
| DeBERTa | **80.8** | **66.1** | **78.9** | 6.6 | **87.1** | **64.3** | **50.5** | **71.2** | **65.4** |
| *1-shot* | | | | | | | | | |
| GPT-3 | 63.7 | 48.2 | 74.0 | 13.6 | 83.0 | 49.5 | 49.2 | 62.5 | 57.8 |
| DeBERTa | **82.1** | **76.1** | **84.2** | **15.6** | **87.4** | **64.1** | **50.3** | **69.6** | **68.4** |
| *few-shot* | | | | | | | | | |
| GPT-3 | 64.1 | 69.6 | 77.0 | **20.8** | 83.1 | 50.9 | **53.0** | 49.0 | 60.0 |
| DeBERTa | **82.1** | **75.0** | **90.4** | 16.9 | **87.4** | **62.2** | 50.8 | **75.0** | **69.6** |

## 4.2 Language modeling, Winograd-style and text completion tasks

The tasks in this category are defined in a familiar language-modeling form and focus on particularly difficult cases of language modeling, cases that involve commonsense reasoning, language understanding, and intricate coreference resolution.

In this group, we consider four NLP tasks: HellaSwag is a text completion task where a language model has to choose the most appropriate multi-word ending, the examples are adversarially filtered to be difficult for language models but easy for humans (Zellers et al., 2019). StoryCloze consists of five-sentence-long stories, the goal is to select the best final sentence based on commonsense knowledge (Mostafazadeh et al., 2016). Winograd is a language-modeling formulation of the WSC task from SuperGLUE (Levesque et al., 2012). WinoGrande is similar to Winograd in its form, but is adversarially mined to contain more difficult examples of coreference resolution (Sakaguchi et al., 2020). We do not include the LAMBADA benchmark (Paperno et al., 2016) here because Brown et al. (2020) used an unknown preprocessing step that disallows direct comparison with GPT-3.

**Results** We show the in-context-learning results from this group in Table 2, where we evaluate the 1.5B DeBERTa with a comparable GPT-3 model. The scores showcase consistently stronger performance of the masked language model, similarly to the language understanding tasks. One difference to those tasks is the rate of scaling, which appears to be similar between the two types of language models (Figure 1).

Table 2: **Results of text completion, language modeling and Winograd-style tasks**    All tasks are measured with accuracy, we show the performance of the largest available DeBERTa (1.4 billion parameters) and of a similarly-sized GPT-3 model, the best results are boldfaced.

|  | HellaSwag | StoryCloze | Winograd | Winogrande | Average |
|---|---|---|---|---|---|
| *0-shot* | | | | | |
| GPT-3 | 54.7 | 73.4 | **76.9** | 58.7 | 65.5 |
| DeBERTa | **62.0** | **83.6** | 74.0 | **61.0** | **70.2** |
| *1-shot* | | | | | |
| GPT-3 | 53.5 | 74.2 | 76.9 | 59.1 | 65.9 |
| DeBERTa | **62.4** | **84.6** | **80.7** | **63.6** | **72.8** |
| *few-shot* | | | | | |
| GPT-3 | 54.9 | 76.1 | 76.9 | 59.1 | 64.8 |
| DeBERTa | **62.5** | **84.8** | **85.6** | **68.8** | **75.4** |

## 4.3   Translation

Translation is a useful benchmark for language models as it evaluates their ability to understand text in one language and produce fluent text in another language. Even though the performance on the translation tasks is arguably very dependent on the composition of training data (especially when we are concerned with monolingual English models), we include translation to demonstrate the generative performance of masked language models.

To directly compare DeBERTa with GPT-3, we use the same SacreBLEU metric (Post, 2018) and the same bitexts. Thus, even though there are more recent (and arguably more thought-out) datasets, we use the French–English pair from the outdated 2014 shared task at the Workshop on Statistical Machine Translation (WMT14; Bojar et al., 2014), and also the Romanian–English and German–English pairs from the WMT16 workshop (Bojar et al., 2016). Our approach differs only in using a different prompt template, as we had to opt for the prompt from Garcia et al. (2023) to get consistent translations: "{$source_language}: {$source_text}\\n {$target_language}: {$target_text}".

**Results**    The SacreBLEU scores on each language pair are given in Table 3. Unlike in the previous two task groups, the tables have turned, and the causal language model clearly outperforms the masked model in all comparisons. We believe that the subpar performance of DeBERTa can be (at least) in part explained by its relatively small and clean monolingual training corpus (Section 3), because the performance on this task is highly dependent on the presence of multilingual data in the corpus (Lin et al., 2022). The rate of improved translation performance with larger scale appears to be similar between the two models (Figure 1).

Table 3: **Machine translation results**    We report SacreBLEU scores (Post, 2018) with signature `BLEU+case.mixed+numrefs.1+smooth.exp+tok.intl+version.1.2.20` (higher is better). The table shows the performance of the largest available DeBERTa (1.4 billion parameters) and of a similarly-sized GPT-3 model, the best results are boldfaced.

|  | DE→EN | EN→DE | FR→EN | EN→FR | RO→EN | EN→RO | Average |
|---|---|---|---|---|---|---|---|
| *0-shot* | | | | | | | |
| GPT-3 | **3.6** | **2.4** | **3.6** | **2.8** | **3.6** | **3.1** | **3.2** |
| DeBERTa | 2.4 | 1.6 | 1.7 | 0.3 | 1.7 | 0.1 | 1.3 |
| *1-shot* | | | | | | | |
| GPT-3 | **25.8** | **13.4** | **27.0** | **19.3** | **26.8** | **10.3** | **18.8** |
| DeBERTa | 23.7 | 5.4 | 23.5 | 9.7 | 17.7 | 2.5 | 13.8 |
| *few-shot* | | | | | | | |
| GPT-3 | **30.5** | **17.7** | **32.2** | **26.1** | **30.1** | **12.9** | **24.9** |
| DeBERTa | 25.1 | 6.6 | 24.5 | 10.8 | 18.9 | 4.1 | 15.0 |

## 4.4 Closed-book question answering and commonsense reasoning

An important quality of modern-day large language models is their ability to learn and retrieve world knowledge, and to have a degree of common sense. The final group of tasks attempts to evaluate these two qualities.

This category of tasks consists of seven datasets in total: Natural Questions (NQs; Kwiatkowski et al., 2019) and Web Questions (WebQs; Berant et al., 2013) are closed-book question-answering datasets sourced from natural web queries; while the original datasets are accompanied by relevant articles that contain the answer, we only ask models a question and then evaluate the exact-match accuracy of their answers. TriviaQA is a very similar dataset, but based on online quizzes (Joshi et al., 2017). The next four tasks fall more into a subcategory of commonsense reasoning datasets. The Physical Interaction: Question Answering dataset evaluates how well a language model is grounded in the real physical world (PIQA; Bisk et al., 2020). The AI2 Reasoning Challenge is a dataset sourced from grade-school science questions that evaluates knowledge and reasoning abilities; this task is divided into ARC-Easy and ARC-Challenge splits, based on their difficulty (Clark et al., 2018). Finally, OpenBookQA evaluates the understanding of common knowledge (Mihaylov et al., 2018).

**Results**    The question-answering performance is given in Table 4. Apparently, the results of DeBERTa are substantially worse on closed-book question answering compared to GPT-3. We believe that this highlights a more general disadvantage of the MLM training objective – the model can often retrieve world knowledge from the rich bidirectional context during training, not needing to store it in its learned weights; similar effect has been shown in retrieval-augmented language models (Samuel et al., 2024). However, the commonsense reasoning abilities are comparable between the two models. The scaling behavior is again similar between the two models (Figure 1). The same is also true about the improvement when given more in-context examples, which are especially important for the tasks evaluated with exact-match accuracy, where the goal is not only to answer correctly but also to match the expected style and form of the gold answers (Figure 4).

Table 4: **Closed-book question answering and commonsense reasoning**    The first three tasks are measured with the exact-match accuracy and the rest is measured with classification accuracy. The table shows the performance of the largest available DeBERTa (1.4 billion parameters) and of a similarly-sized GPT-3 model, the best results are boldfaced. A detailed description of the evaluation method is given in Appendix E, full results are in Appendix F.

|  | NQs | TriviaQA | WebQs | PIQA | ARC-C | ARC-E | Open-BookQA | Average |
|---|---|---|---|---|---|---|---|---|
| *0-shot* | | | | | | | | |
| GPT-3 | **4.4** | **19.7** | **4.6** | **75.1** | 35.5 | 53.8 | **46.8** | **34.4** |
| DeBERTa | 0.8 | 6.9 | 1.5 | 72.9 | **36.5** | **55.1** | 45.8 | 31.4 |
| *1-shot* | | | | | | | | |
| GPT-3 | **5.4** | **26.5** | **9.2** | 74.4 | 36.4 | **55.9** | **46.4** | **36.3** |
| DeBERTa | 2.6 | 14.3 | 5.1 | 73.0 | **37.1** | 55.1 | 45.7 | 33.3 |
| *few-shot* | | | | | | | | |
| GPT-3 | **9.7** | **32.1** | **19.6** | 74.3 | 36.7 | **59.1** | **50.6** | **40.3** |
| DeBERTa | 4.4 | 17.9 | 9.9 | **74.5** | **39.6** | 57.7 | 50.4 | 36.3 |

## 5 Related work

**Few-shot finetuning with masked language models**    While our work demonstrates the emergence of in-context learning in masked language models, prior research has explored different approaches to few-shot learning with these architectures. The dominant paradigm has been few-shot *finetuning*, where the model's weights are updated using a small number of examples. Studies by Schick and Schütze (2021), Gao et al. (2021), and Xia et al. (2022) showed promising results with this approach. However, these methods require additional training steps with a complicated training objective,

making them more complex to implement compared to the simple prompting-based in-context learning demonstrated in our work. Despite the generally lower performance of in-context learning compared to few-shot finetuning (Liu et al., 2022), its simplicity and immediacy have made it the preferred choice in many practical applications.

**Other large masked language models**   Our choice of DeBERTa for this study was motivated by its unique combination of size and capability to handle extended context lengths. While larger masked language models exist, such as Megatron BERT with 3.9 billion parameters (unfortunately not publicly available; Shoeybi et al., 2019) and XLM-RoBERTa with 10.7 billion parameters (Goyal et al., 2021), they have limitations that make them less suitable for studying in-context learning. Megatron BERT lacks mechanisms for length generalization, which is crucial for processing long prompts with multiple examples, while XLM-RoBERTa's multilingual nature and restricted sequence length of 512 tokens would confound our analysis. DeBERTa's architecture, particularly its relative positional embeddings, makes it an ideal candidate for exploring how masked language models scale with in-context learning.

**Hybrid masked-causal models**   Our empirical findings, particularly the complementary strengths of masked and causal models demonstrated in Section 4, suggest significant potential in combining these approaches. Several architectures have already explored this direction, even if inadvertently: T5 (Raffel et al., 2020), BART (Lewis et al., 2020) and GLM (Du et al., 2022) introduced autoregressive fill-in-the-blank objectives; CM3 developed a causal-mask approach (Aghajanyan et al., 2022); and PrefixLM implemented a partially bidirectional causal model (Dong et al., 2019; Raffel et al., 2020). These efforts align with our observations about the distinct advantages of masked and causal objectives. The recent work by Ding et al. (2024) provides theoretical support for this direction, demonstrating that prefix language models, which combine aspects of both architectures, are particularly well-suited for in-context learning.

## 6   Conclusion

This paper demonstrates that masked language models can be capable in-context learners. We show that these models – often considered deprecated and limited only to finetuning – can match and sometimes even exceed the performance of their causal counterparts in this domain. Our evaluation reveals that masked and causal models exhibit remarkably similar characteristics in terms of overall performance, scaling behavior, and improvements with additional in-context demonstrations. Most notably, we validate these capabilities using DeBERTa without any architectural modifications or additional training. We achieve this through carefully designed inference methods that unlock the model's latent generative abilities.

Our findings point to several promising directions for future research. First, DeBERTa's performance could likely be enhanced through straightforward improvements such as training on larger and more diverse corpora, increasing model scale, and extending the pretraining context length. More fundamentally, the complementary strengths we observed between masked and causal models – where each architecture excels in different tasks – suggest an exciting opportunity to develop hybrid approaches that combine the best of both paradigms. Rather than viewing these as competing architectures, future work might explore how to synthesize their distinct advantages into more capable and versatile language models.

These results argue for a broader reconsideration of how we approach language model architecture and training. The field's recent focus on causal models, while productive, may have prematurely discounted the potential of alternative approaches that are not limited to unidirectional text processing. Our work demonstrates that the path forward likely involves embracing architectural diversity rather than converging on a single dominant paradigm.

## Acknowledgments and Disclosure of Funding

I am deeply grateful to Lilja Øvrelid, Andrey Kutuzov, and Erik Velldal for providing insightful feedback, for their never-ending encouragement and support, and for making Oslo a warm and welcoming place.

This work is fully funded by the University of Oslo. The computations were performed on resources provided through Sigma2 – the national research infrastructure provider for high-performance computing and large-scale data storage in Norway. We acknowledge Norway and Sigma2 for awarding this project access to the LUMI supercomputer, owned by the EuroHPC Joint Undertaking, hosted by CSC (Finland) and the LUMI consortium through project 5000144.

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

# A Examples of text generation

To also give a sense of the quality of the text produced by DeBERTa, we include some examples of text generation in this section. We use the exact same example prompts that were used in the GPT-3 paper (Brown et al., 2020), to provide a fair estimate of the generative qualities. All text completions were generated with nucleus sampling (Holtzman et al., 2020).[5] We compare the largest DeBERTa 1.5B with OPT 1.3B, an openly available replication of the GPT-3 1.3B (Zhang et al., 2022).

## A.1 Learning and using novel words

Based on studies in developmental linguistics (Carey and Bartlett, 1978), this task tests the ability to understand and productively use new words; specifically using a word in a sentence after seeing it defined only once. We qualitatively test this ability in a generative one-shot setting, using the prompts provided below – there, the human-provided prompts are rendered as normal text while the generated completions are rendered in boldface. The prompts are taken from Brown et al. (2020, Section 3.9.5).

**Results** Overall, DeBERTa provides more appropriate example sentences than OPT. While the 'farduddle' and 'screeg' sentences from DeBERTa are not very descriptive, the rest of sentences are informative and fitting the word definitions. Note how the model tried to invent a plural inflection of 'yalubalu', the suffix '-a' is morphologically plausible, but the stem is fumbled, possibly because of subword tokenization. The examples generated by OPT are of lesser quality; it either repeats the definition (in 'farduddle'), repeats the one-shot example (in 'yalubalu') or provides an unfitting example (in 'screeg').

```
1   A "whatpu" is a small, furry animal native to Tanzania. An example of a sentence
    ↪   that uses the word whatpu is:
2   We were traveling in Africa and we saw these very cute whatpus.
3
```

```
4   To do a "farduddle" means to jump up and down really fast. An example of a sentence
    ↪   that uses the word farduddle is:
```
DeBERTa:
```
5   I did a farduddle.
```
OPT:
```
5   I farduddle jumped up and down really fast.
```

```
4   A "yalubalu" is a type of vegetable that looks like a big pumpkin. An example of a
    ↪   sentence that uses the word yalubalu is:
```
DeBERTa:
```
5   We were in the market and we saw these very big yambolula.
```
OPT:
```
5   We were traveling in Africa and we saw these yalubalu.
```

```
4   A "Burringo" is a car with very fast acceleration. An example of a sentence that
    ↪   uses the word Burringo is:
```
DeBERTa:
```
5   We were driving a very fast Burringo.
```
OPT:
```
5   We saw Burringos all over Africa on this trip.
```

```
4   A "Gigamuru" is a type of Japanese musical instrument. An example of a sentence that
    ↪   uses the word Gigamuru is:
```
DeBERTa:
```
5   A Japanese musician, who plays the Gigamuru, is very famous.
```
OPT:
```
5   I was walking down the street and I saw this very cute gigamuru.
```

---

[5]Using these hyperparameters: `top_k=64`, `top_p=0.9`, `temperature=0.2`.

```
 4   To "screeg" something is to swing a sword at it. An example of a sentence that uses
     ↪  the word screeg is:
```
**DeBERTa:**
```
 5   A sword-screeg.
```

**OPT:**
```
 5   I was screeging at the computer.
```

## A.2   Correcting English grammar

English grammar correction is another task used for qualitative evaluation of GPT-3. Provided three demonstrations, the model is given an incorrect English sentence and is tasked to correct it. The prompts are taken from Brown et al. (2020, Section 3.9.6) and are shown below.

**Results**   While the corrections are not perfect, DeBERTa outputs more sensible completions compared to OPT, similarly to the previous task. A notable correction is in the second example, where DeBERTa tried to improve the understandability, not only grammar.

```
 1   Poor English input: I eated the purple berries.
 2   Good English output: I ate the purple berries.
 3   Poor English input: Thank you for picking me as your designer. I'd appreciate it.
 4   Good English output: Thank you for choosing me as your designer. I appreciate it.
 5   Poor English input: The mentioned changes have done.
 6   Good English output: The requested changes have been made.
```

```
 7   Poor English input: I'd be more than happy to work with you in another project.
```
**DeBERTa:**
```
 8   Good English output: I'd be more than happy to work with you in another project.
```

**OPT:**
```
 8   Good English output: I'd be more than happy to work with you in another project.
```

```
 7   Poor English input: Please provide me with a short brief of the design you're
     ↪  looking for and that'd be nice if you could share some examples or project you
     ↪  did before.
```
**DeBERTa:**
```
 8   Good English output: Please provide me with a brief of the design you are looking for
     ↪  and some examples of your previous work.
```

**OPT:**
```
 8   Good English output: I'll provide you with a brief of the design you're looking for
     ↪  and that'd be nice if you could share some examples or project you did before.
```

```
 7   Poor English input: The patient was died.
```
**DeBERTa:**
```
 8   Good English output: The patient has died.
```

**OPT:**
```
 8   Good English output: The patient was revived.
```

```
 7   Poor English input: We think that Leslie likes ourselves.
```
**DeBERTa:**
```
 8   Good English output: We think that we like ourselves.
```

**OPT:**
```
 8   Good English output: We think that Leslie likes herself.
```

```
 7   Poor English input: Janet broke Bill on the finger.
```
**DeBERTa:**
```
 8   Good English output:Janet broke Bill's finger.
```

**OPT:**
```
 8   Good English output: Janet broke Bill on the finger.
```

```
       7   Poor English input: Mary arranged for, in St. Louis, John to rent a house cheap.
DeBERTa:  8   Good English output: Mary arranged for, in St. Louis, John to rent a house.

   OPT:   8   Good English output: John rented a house cheap.
```

## B   Ablation study of text generation

We empirically evaluate different approaches for text generation with DeBERTa language models. In particular, we study how many additional mask tokens should be used during autoregressive generation, and we also compare our approach with previously proposed methods based on Markov-chain Monte Carlo sampling (Wang and Cho, 2019). For this analysis, we use German-to-English machine translation as a representative generative task. We evaluate different generation methods using one-shot prompts (following Section 2.1) and the largest 1.5B DeBERTa model. The translation quality is measured with SacreBLEU score, using the same signature as in the main experiments.

**Results**   The results shown in Table 5 demonstrate that using additional mask tokens during generation substantially improves the performance of DeBERTa. This aligns with our hypothesis that additional masks help to reduce the effect of the end-of-sequence token on the generated text. The results also show that while adding a fourth mask token still marginally improves the performance, the gain is negligible compared to using three mask tokens.

We also compare our autoregressive approach with the Gibbs sampling method proposed by Wang and Cho (2019). There we relied on the same hyperparameters that are suggested in the official implementation: 500 total iterations, 250 burn-in iterations with top-100 sampling and temperature of 1.0.[6] Their sampling-based approach performs substantially worse than autoregressive generation, regardless of the initialization strategy. We noticed that it often produces infinite token repetitions or locally-coherent but globally-disconnected pieces of text.

Table 5: **Ablation study of different generation methods applied to DeBERTa**   Evaluated using one-shot setting and the largest DeBERTa 1.5B model on German-to-English translation with SacreBLEU score.

| Generative method | DE→EN |
|---|---|
| Autoregressive generation; 1 mask | 10.0 |
| Autoregressive generation; 2 masks | 22.4 |
| Autoregressive generation; 3 masks (our method) | 23.7 |
| Autoregressive generation; 4 masks | **23.9** |
| Markov-chain Monte-Carlo (random initialization; Wang and Cho, 2019) | 1.6 |
| Markov-chain Monte-Carlo (mask initialization; Wang and Cho, 2019) | 2.6 |

## C   Ablation study of ranking implementation

We mentioned some drawbacks of calculating the pseudo-log-likelihood score (PLL) as per Salazar et al. (2020), and how we mitigate these problems, in Section 2.2. This section supports our decision with a quantitative analysis of different ranking approaches. We test them on the ReCoRD task from SuperGLUE (Zhang et al., 2018; Wang et al., 2019), where the goal is to rank different named entities based on their appropriatness. Since the problem of the original PLL is in estimating likelihoods of long multi-loken expressions, we choose this task to highlight the differences. An example of a

---

[6]Available on GitHub: https://github.com/nyu-dl/bert-gen.

prompt-formatted sample from ReCoRD is given below, the possible completions (all of which are named entities) are boldfaced:

```
1   Suspended hundreds of feet in the air amid glistening pillars of ice illuminated
    ↪   with ghostly lights from below, this could easily be a computer-generated scene
    ↪   from the latest sci-fi blockbuster movie. But in fact these ethereal photographs
    ↪   were taken in real life, and show extreme sportsman and climber Stephan
    ↪   Siegrist, 43, ascending the Voringsfossen icefall which is part of a gigantic
    ↪   glacier in Eidfjord, Norway. The stunning images were captured by fellow
    ↪   mountaineer and photographer Thomas Sanf. While the 500ft frozen waterfall is
    ↪   regularly scaled by climbers during daylight, he said he wanted to capture the
    ↪   beauty of the falls by night.
2   - Stunning images captured by photographer Thomas Sanf as climber Stephan Siegrist,
    ↪   43, scaled frozen waterfall
3   - The Voringsfossen fall is liquid for most of the year, but in winter freezes into
    ↪   a 500ft cliff favoured by climbers
4   - Hundreds of adventurers attempt the climb by day, but very few attempt the ascent
    ↪   at night, as pictured here
5   - With bright lights illuminating his efforts from below, Mr {$answer:-Stephan
    ↪   Siegrist/Voringsfossen/Eidfjord/Norway/Thomas Sanf} appears to be on the set of
    ↪   a sci-fi movie
```

**Previous work**   After running the initial experiments, we were informed about the related work by Kauf and Ivanova (2023). Their study addresses a similar problem of the naive pseudo-log-likelihood scoring as this paper, but they do not target strongly correlated contiguous tokens in general, they focus on words that are split into multiple tokens – as they observed that PLL overestimates the likelihood of those sequences. We include their two proposed solutions in the comparison of different ranking methods.

**Results**   In Table 6, we compare the original implementation of PLL by Salazar et al. (2020) that only masks the target subword; our approach that also masks next two subwords; other two alternatives that mask two or four subwords in total; the PLL-word-l2r and PLL-whole-word scoring functions by Kauf and Ivanova (2023); and the exact unidirectional computation of log-likelihood (using the same input formatting as for generation). The additional masks clearly help to make better estimates while the exact computation seems to not be appropriate for inherently bidirectional models.

Table 6: **Ablation study of different ranking methods applied to DeBERTa**   Evaluated using zero-shot setting and the largest DeBERTa 1.5B model on ReCoRD.

| Ranking method | ReCoRD (EM) | ReCoRD ($F_1$) |
|---|---|---|
| Pseudo log-likelihood; 1 mask (Salazar et al., 2020) | 80.9 | 81.6 |
| Pseudo log-likelihood; 2 masks | 86.0 | 86.8 |
| Pseudo log-likelihood; 3 masks (our method) | **87.1** | **87.9** |
| Pseudo log-likelihood; 4 masks | 86.9 | 87.8 |
| Pseudo log-likelihood; word–l2r (Kauf and Ivanova, 2023) | 78.2 | 78.8 |
| Pseudo log-likelihood; whole–word (Kauf and Ivanova, 2023) | 75.8 | 76.5 |
| Exact log-likelihood (unidirectional) | 77.2 | 77.8 |

# D   Detailed SuperGLUE results

This appendix provides a detailed analysis of DeBERTa's performance on the SuperGLUE benchmark, focusing on two aspects: the relationship between the number of demonstration examples and model performance (Figure 5), and the task-specific scaling behavior (Figure 6).

**Effect of shot count**   Figure 5 demonstrates how DeBERTa's performance changes as we increase the number of in-context examples from 0 to 32. Unlike the main results where we optimize the number of shots for each subtask independently, here we deliberately use the same number of shots across all tasks to provide a controlled comparison with GPT-3's results from Brown et al. (2020); there, SuperGLUE tasks are the only ones with such detailed few-shot evaluation.

The plot reveals several interesting patterns:

1. The steepest improvement occurs between 0 and 4 shots, suggesting that even a small number of examples provides substantial benefit.

2. The overall trend is very similar between DeBERTa and GPT-3, which again shows that masked language models can be just as good in-context learners as causal language models.

3. The performance of DeBERTa decreases after 8 or more shots. We believe that this is mainly caused by its imperfect processing of contexts longer than 512 tokens – as discussed in Section 2.3.

Figure 5: **The average performance on the SuperGLUE benchmarks as a function of number of shots**   As opposed to the other SuperGLUE *few*-shot results, where we select the number of shots for each subtask according to the performance on its training split, here all subtasks are evaluated with the same number of shots. In this way, we can compare DeBERTa 1.5B directly to Figure 3.8 from Brown et al. (2020), which gives the same evaluation for GPT 175B (unfortunately not for smaller, more comparable, models).

**Task-specific scaling analysis**   Figure 6 breaks down the scaling behavior for each SuperGLUE task individually, revealing the varying impacts of model size across different language understanding challenges. The plots demonstrate that while DeBERTa and GPT-3 both exhibit generally positive scaling trends, their scaling patterns differ substantially across task types. Most notably, DeBERTa shows steeper scaling curves than GPT-3 on the majority of tasks. This suggests that bidirectional attention mechanisms may provide particular advantages for scaling some language understanding capabilities.

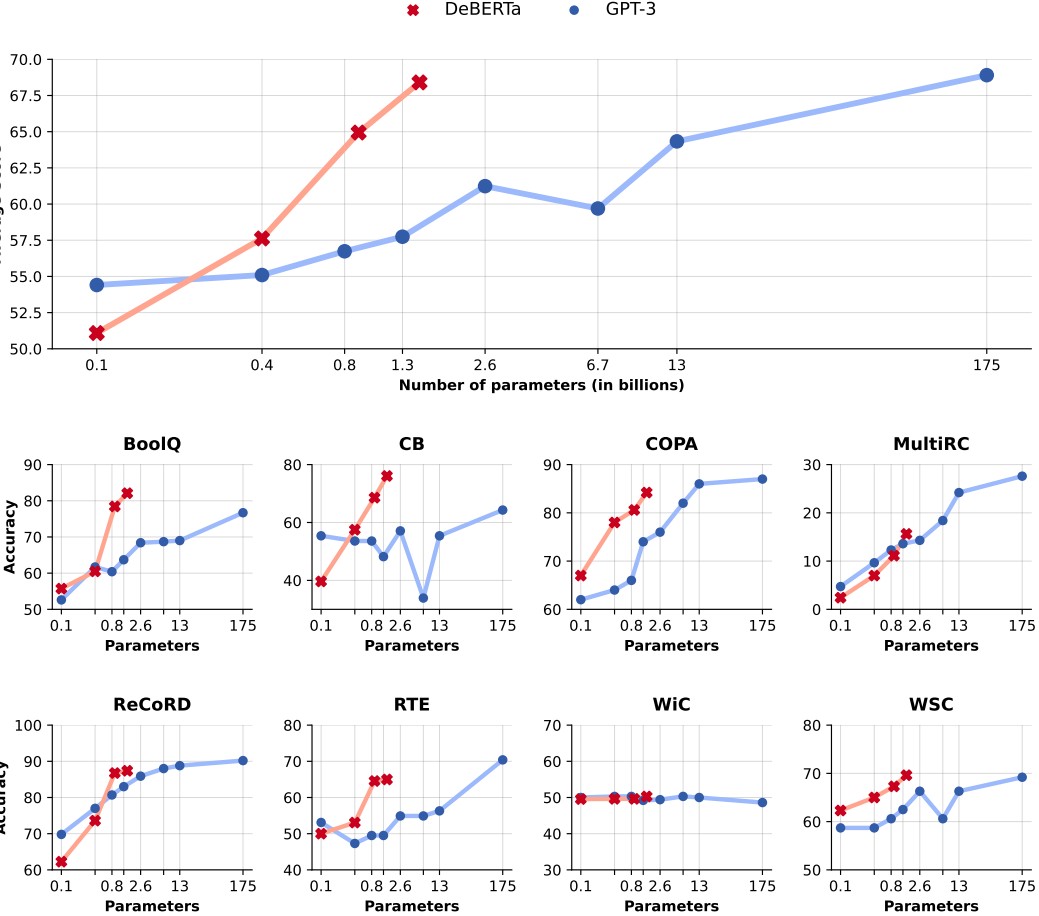

Figure 6: **Detailed evaluation of natural language understanding**    As opposed to the overall results presented in Figure 1, these plots show the scaling behavior on each subtask of the SuperGLUE benchmark. We can see the DeBERTa out-scales GPT-3 consistently across all tasks, with the exception of WiC where both models fail to show any learning ability.

# E   Evaluation details

This appendix provides more details about our evaluation setup to make it easier to reproduce our results; the source code is available at `https://github.com/ltgoslo/bert-in-context`. In general, we follow Brown et al. (2020) in all decisions about the implementation.

**Newline separator**    Note we use newlines in the displayed prompts for better readability, but we do not actually use them as the DeBERTa tokenizers cannot encode a newline character (they convert it to the standard whitespace character instead). Instead, we convert all newline characters to double-escaped '`\\n`  ' string with a whitespace character, which then acts as paragraph/information separator.

**Few-shot prompting**    For each evaluated sample, the example demonstrations are randomly selected (without replacement) from the training set of each task; if the training set is not available, we sample from the only available dataset split, making sure not to select the same sample as the evaluated

one. We format these examples using the respective prompt templates and concatenate them together, joined by two newline characters. The numbers of shots used for each task are given in Appendix F.

## E.1 Needle in a haystack

The needle is a randomly generated 6-digit number (from $100\,000$ to $999\,999$). The prediction is produced via constrained generation that only allows sequences of tokens that form 6-digit numbers. We consider a prediction to be correct only if it exactly matches the needle, which means that a trivial baseline has the accuracy of $1/900\,000$.

The evaluated models are prompted with the template shown below. Similarly to Hsieh et al. (2024), we use Paul Graham's essays to fill the context (`$prefix_lines` and `$suffix_lines`).[7] The essays are sentence-segmented and concatenated to fill the desired total sequence length.

```
1   > Some special magic number is hidden within the following articles. Make sure to
    ↪   memorize it. I will quiz you about the magic number afterwards.
2
3   {$prefix_lines}
4   The magic number is {$needle}.
5   {$suffix_lines}
6
7   > Question: What is the special magic number mentioned in the provided text?
8   > Answer: The special magic number mentioned in the provided text is
```

## E.2 Language understanding

This section provides the prompts used for the eight SuperGLUE tasks, all prompt templates are taken from the GPT-3 evaluation setup.

**BoolQ**  This task is evaluated by ranking texts formatted as shown below with two possible `$answer`s, **yes** or **no**.

```
1   {$passage}
2   question: {$question}?
3   answer: {$answer:-yes/no}
```

**CB**  Evaluated by ranking texts formatted as shown below with three possible `$answer`s, **true**, **false** or **neither**.

```
1   {$premise}
2   question: {$hypothesis}; true, false, or neither?
3   answer: {$answer:-true/false/neither}
```

**COPA**  We rank two possible substitutions of `$answer` that follow the premise. `$connector` is formatted based on the question type: 'because' if the type is 'cause', otherwise 'therefore' is used.

```
1   {$premise} {$connector:-because/therefore} {$answer}
```

---

[7]Available online at https://www.paulgraham.com/articles.html

**MultiRC**    The potential answer is substituted for `$option` and then we rank two possible substitutions for `$answer`: `[True]` or `[False]`.

```
1    READING COMPREHENSION ANSWER KEY
2
3    {$paragraph}
4
5    {$question}
6    - {$answer:-[True]/[False]} {$option}
```

**ReCoRD**    Here we rank the possible entity names as substitutions for `$answer`. Note that `$paragraph` often includes summaries that, in a way, act as few-shot examples (see the formatted zero-shot example in Appendix C).

```
1    {$passage}
2    - {$answer_prefix}{$answer}{$answer_suffix}
```

**RTE**    Ranking of two possible completions in this binary classification task: `True` or `False`.

```
1    {$premise}
2    question: {$hypothesis} True or False?
3    answer: {$answer:-True/False}
```

**WiC**    Another binary task with two possible substitutions: `yes` or `no`. Note that this prompt template was not working for any GPT-3 models and it is also not working for the models evaluated in this paper, all models are just randomly guessing the answers (Table 1).

```
1    {$sentence1}
2    {$sentence2}
3    question: Is the word '{$word}' used in the same way in the two sentences above?
4    answer: {$answer:-yes/no}
```

**WSC**    We rank possible substitutions for `$answer`.

```
1    Final Exam with Answer Key
2    Instructions: Please carefully read the following passages. For each passage, you
     ↪  must identify which noun the pronoun marked in *bold* refers to.
3    =====
4
5    Passage: {$text}
6    Question: In the passage above, what does the pronoun "*{$span_text}*" refer to?
7    Answer: {$answer}
```

### E.3 Language modeling

This group of tasks uses very straightforward prompt templates as all of these tasks are different variants of text completion.

**HellaSwag**   Here, the task is to select the most likely completion (**$answer**) that follows after **$context**.

```
1    {$activity_label}: {$context}{$answer}
```

**StoryCloze**   The goal is to select the most suitable **$answer** that completes a story laid out by four previous sentences.

```
1    {$sentence_1} {$sentence_2} {$sentence_3} {$sentence_4} {$answer}
```

**Winograd**   **$answer** should be substituted by the correct entity (coreference resolution).

```
1    {$context_prefix} {$answer} {$context_suffix}
```

**Winogrande**   Same as Winograd:

```
1    {$context_prefix} {$answer} {$context_suffix}
```

### E.4 Translation

All language pairs and all evaluation setups (zero-shot, one-shot and few-shot) use the same prompt template given below. This is the only time when we decided to differ from the GPT-3 setup as its very simple (and non-informative) prompt was not working for one-shot evaluation of DeBERTa models. The models are then asked to complete the prompt – using beam search decoding with 4 beams and the default HuggingFace hyperparameters[8] – and the generation is stopped after producing the special newline character **\\n**.

```
1    {$source_language}: {$source_text}
2    {$target_language}:
```

For reference, here are the two prompt templates used for GPT-3 (for zero-shot and for one/few-shot, respectively):

```
1    Q: What is the {$target_language} translation of {$source_text}
2    A:
```

```
1    {$source_text} =
2
```

---

[8]https://huggingface.co/docs/transformers/v4.41.3/en/main_classes/text_generation#transformers.GenerationMixin.generate

### E.5 Closed-book question answering

This group of tasks mixes two types of in-context evaluation: text generation (Natural Questions, TriviaQA and Web Questions) and text ranking (PIQA, ARC and OpenBookQA). The prompt setup exactly follows GPT-3.

**Natural Questions**    Here, the goal is to generate an answer based on `$question`.

```
1   Q: {$question}
2   A:
```

**TriviaQA**    Same as Natural Questions:

```
1   Q: {$question}
2   A:
```

**Web Questions**    Same as Natural Questions:

```
1   Q: {$question}
2   A:
```

**PIQA**    Here, the goal is to select the most suitable text completion by substituting for `$answer`.

```
1   {$goal} {$answer}
```

**ARC (challenge) and ARC (easy)**    This a multiple choice test, the correct `$answer` has to be selected.

```
1   Question: {$question}
2   Answer: {$answer}
```

**OpenBookQA**    Similarly, the goal is to select the correct `$answer`.

```
1   {$question} {$answer}
```

# F  All results

For reference, we list all results of the DeBERTa models evaluated throughout this paper in Table 7. The GPT-3 results that were used for comparison are published in Brown et al. (2020, Table H.1).

Table 7: **Results of all evaluations performed in this paper**  The second and third column shots the dataset splits and evaluation metrics, both of them replicating the GPT-3 evaluation setup. Note the BLEU scores used for evaluating translation are SacreBLEU scores with signature BLEU+case.mixed+numrefs.1+smooth.exp+tok.intl+version.1.2.20.

| Task | Split | Metric | $n$ shots | 0-shot (1.5B) | 1-shot (0.1B) | 1-shot (0.4B) | 1-shot (0.9B) | 1-shot (1.5B) | $n$-shot (1.5B) |
|---|---|---|---|---|---|---|---|---|---|
| BoolQ | dev | acc. | 4 | 80.8 | 55.7 | 60.5 | 78.4 | 82.1 | 82.1 |
| CB | dev | acc. | 4 | 66.1 | 39.6 | 57.5 | 68.6 | 76.1 | 75.0 |
| CB | dev | $F_1$ | 4 | 46.1 | 23.8 | 39.8 | 47.1 | 57.0 | 57.6 |
| COPA | dev | acc. | 64 | 78.9 | 67.0 | 78.0 | 80.6 | 84.2 | 90.4 |
| MultiRC | dev | EM acc. | 4 | 6.6 | 2.4 | 7.0 | 11.1 | 15.6 | 16.9 |
| MultiRC | dev | $F_{1\alpha}$ | 4 | 61.6 | 57.2 | 57.4 | 57.0 | 67.9 | 69.2 |
| ReCoRD | dev | EM acc. | 4 | 87.1 | 62.3 | 73.6 | 86.8 | 87.4 | 87.4 |
| ReCoRD | dev | $F_1$ | 4 | 87.9 | 63.0 | 74.3 | 87.5 | 88.1 | 88.2 |
| RTE | dev | acc. | 8 | 64.3 | 50 | 53.1 | 64.5 | 65.0 | 62.2 |
| WiC | dev | acc. | 16 | 50.5 | 49.6 | 49.6 | 49.6 | 50.3 | 50.2 |
| WSC | dev | acc. | 16 | 71.2 | 62.3 | 65.0 | 67.3 | 69.6 | 75.0 |
| **Average** | — | — | — | 65.4 | 51.1 | 57.6 | 64.9 | 68.4 | 69.6 |
| HellaSwag | dev | acc. | 16 | 62.0 | 36.9 | 51.3 | 58.7 | 62.4 | 62.5 |
| StoryCloze | test | acc. | 32 | 83.6 | 69.5 | 77.0 | 82.4 | 84.6 | 84.8 |
| Winograd | test | acc. | 32 | 74.0 | 59.3 | 68,1 | 76.2 | 80.7 | 85.6 |
| Winogrande | dev | acc. | 32 | 61.0 | 49.8 | 54.8 | 60.6 | 63.6 | 68.8 |
| **Average** | — | — | — | 70.2 | 53.9 | 62.8 | 69.5 | 72.8 | 75.4 |
| DE–EN | test | BLEU | 16 | 2.4 | 0.2 | 4.6 | 20.0 | 23.7 | 25.1 |
| EN–DE | test | BLEU | 16 | 1.6 | 0.2 | 0.4 | 3.4 | 5.4 | 6.6 |
| FR–EN | test | BLEU | 16 | 1.7 | 0.2 | 8.7 | 21.9 | 23.5 | 24.5 |
| EN–FR | test | BLEU | 16 | 0.3 | 0.0 | 0.3 | 4.6 | 9.7 | 10.8 |
| RO–EN | test | BLEU | 16 | 1.7 | 0.2 | 4.3 | 16.0 | 17.7 | 18.9 |
| EN–RO | test | BLEU | 16 | 0.1 | 0.0 | 0.2 | 1.2 | 2.5 | 4.1 |
| **Average** | — | — | — | 1.3 | 0.1 | 3.1 | 11.2 | 13.8 | 15.0 |
| Natural Questions | test | EM acc. | 16 | 0.8 | 0.1 | 0.6 | 2.1 | 2.6 | 4.4 |
| TriviaQA (wiki) | dev | EM acc. | 16 | 6.9 | 0.9 | 3.8 | 13.6 | 14.3 | 17.9 |
| Web Questions | test | EM acc. | 32 | 1.5 | 0.3 | 1.0 | 4.5 | 5.1 | 9.9 |
| PIQA | dev | acc. | 32 | 72.9 | 62.4 | 69.6 | 71.6 | 73.0 | 74.5 |
| ARC (challenge) | test | acc. | 32 | 36.5 | 25.3 | 33.2 | 35.9 | 37.1 | 39.6 |
| ARC (easy) | test | acc. | 32 | 55.1 | 39.6 | 46.3 | 53.3 | 55.1 | 57.7 |
| OpenBookQA | test | acc. | 96 | 45.8 | 35.0 | 41.8 | 42.8 | 46.4 | 50.4 |
| **Average** | — | — | — | 31.4 | 23.2 | 28.0 | 32.0 | 33.3 | 36.3 |

