# OpenReview forum: "BERTs are Generative In-Context Learners"
_NeurIPS.cc/2024/Conference — NeurIPS 2024 poster_

### Official Review · Reviewer_gXpi · 2024-06-29

**Soundness:** 2
**Presentation:** 3
**Contribution:** 2
**Rating:** 7
**Confidence:** 4

**Summary:**

This paper proposes a finding, indicating that masked language models like BERT can be used as in-context learners. This work suggests that there is potential for a hybrid training, where masked/causal language models can be involved, to take advantage of both objectives. The authors show that masked and generative LMs can outperform each other on different categories of tasks.

After reading the authors' response, I think they made some good points and I think I will give accept to this paper.

**Strengths:**

Rethinking and attempting to discover the advantage of masked LM is very impressive, especially in the era of generative large language models, where most researchers put their efforts into improving the generative LLM performance. The figures are pretty clear.

**Weaknesses:**

1. In Figure 1, the authors demonstrate that DeBERTa outperforms GPT-3 in language understanding. But this is more about the natural advantage of the masked language model. The motivations for providing such experimental data are quite vague to me. Additionally, I don't see the advantages of masked models in the other three tasks (Figure 1). In logic, if one method only occasionally shows its advantage, then this method can lack robustness and generalizability. Furthermore, since the masked language models lack large versions (For example 175B), it's quite necessary to justify why use the smaller masked LM over the large and powerful generative LM.

2. This paper mainly uses GPT-3 and DeBERTa to perform experiments. Several issues can weaken this paper. In the era of LLM, GPT-3 can be outdated. There are amount of generative LMs, which are redesigned and well-trained to make them much more lightweight and powerful than GPT-3. Even DeBERTa can demonstrate certain advantages over GPT-3, such experimental results cannot fully support the statement of this paper. In the same way, masked language models have many implementations. The authors need to prove that either DeBERTa is the best masked LM or provide more elements using other masked LMs.

3. As a technical paper, either provide a wide and systemically empirical study or provide a solid mathematical proof.

**Questions:**

As I explained the weaknesses, authors are encouraged to answer the questions I made in the weaknesses section. For details:

1. Which direction do you plan to conduct your research: "either provide a wide and systemically empirical study or provide a solid mathematical proof"? Please specify that and provide the relevant content.
2. At least add Llama-2, Llama-3, Phi-2, Phi-3, Gemma-2B/7B as the exemplar generative LMs. For masked LM, if you cannot prove DeBERTa is the best masked LM (or find literature that supports your model choice), you should at least add BERT, RoBRETa, DistillBERT, T5.

**Limitations:**

1. Lack of experiments to support the statement/conclusion of this paper.
2. Unclear the research conduct path.

---

> ### Author Rebuttal · Authors · 2024-08-07
>
> Thank you very much for your review!
> ____
> > In Figure 1, the authors demonstrate that DeBERTa outperforms GPT-3 in language understanding. But this is more about the natural advantage of the masked language model. The motivations for providing such experimental data are quite vague to me.
>
> Indeed, one contribution of our paper is that we have shown that MLMs have advantage on this kind of tasks. It is not clear to us how this is a weakness. It might seem 'natural' in hindsight, but we are not aware of any prior works that show that the MLM objective is better for learning how to understand language (without the additional influence of finetuning).
> ____
> >  Additionally, I don't see the advantages of masked models in the other three tasks (Figure 1). In logic, if one method only occasionally shows its advantage, then this method can lack robustness and generalizability.
>
> MLMs are also clearly better on the second group of tasks (HellaSwag, StoryCloze, Winograd, Winogrande), see Figure 1 and 2. But we never claimed that MLMs are better overall, we clearly conclude that MLMs should be combined with CLMs during pretraining, to combine their advantages.
> ____
> > Furthermore, since the masked language models lack large versions (For example 175B), it's quite necessary to justify why use the smaller masked LM over the large and powerful generative LM.
>
> We compare language models of comparable size, using the existing pretrained models. It is completely out of budget for us to pretrain a 175B MLM for these experiments. However, we included scaling experiments to demonstrate that MLMs scale just as well as CLMs (Figure 1).
> ____
> > This paper mainly uses GPT-3 and DeBERTa to perform experiments. Several issues can weaken this paper. In the era of LLM, GPT-3 can be outdated. There are amount of generative LMs, which are redesigned and well-trained to make them much more lightweight and powerful than GPT-3.
>
> We chose two models that are comparable, in terms of the release date, size of training data, number of pretraining steps and size (as described in Section 3). As an example, you suggest to include Llama3 in the comparison, but this model is trained on 1000x bigger dataset (approximately), for much more steps and the smallest Llama3 is more than 5x larger than the largest DeBERTa; we are not sure how this additional comparison would help to prove/disprove our claim that MLMs are capable of in-context learning. Note that we are not claiming that the 4-year-old DeBERTa model is the state of the art (even though it is somewhat intriguing that DeBERTa matches Llama3 on tasks such as OpenBookQA or Winogrande).
> ____
> > The authors need to prove that either DeBERTa is the best masked LM or provide more elements using other masked LMs.
>
> Our main claim is that MLMs can function as in-context learners, for this, it is enough to show that one such model exhibits these abilities.  We do not claim that DeBERTa is the best masked language model. There are technical reasons for choosing DeBERTa over other models, we describe them in Related Work as well as in Section 2. In short, we need a model that is capable of processing long inputs (> 512 tokens), that is large (>1B parameters) and that is primarily trained on English (to fairly compare against GPT-3). As far as we know, DeBERTa is the only model that satisfies this.
> ____
> > As I explained the weaknesses, authors are encouraged to answer the questions I made in the weaknesses section. For details: ...
>
> We hope that we addressed these questions with the comments above.

---

> > ### Comment · Reviewer_gXpi · 2024-08-12
> >
> > Dear authors,
> >
> > Thank you for your additional information. I have increased my score, hope you are doing well in this work.

---

### Official Review · Reviewer_7Ges · 2024-07-01

**Soundness:** 3
**Presentation:** 4
**Contribution:** 3
**Rating:** 7
**Confidence:** 4

**Summary:**

This paper argues that masked language models (MLMs) are just as capable as autoregressive language models at in-context learning. To demonstrate this, the authors propose a generative inference technique that allows DeBERTa to generate text, as well as a hybrid autoregressive/MLM pseudo-log-likelihood estimation technique that allows DeBERTa to rank text sequences by likelihood.

DeBERTa and GPT-3 models are compared on a series of NLP benchmarking tasks in an in-context learning setup. While GPT-3 excels at translation and QA, DeBERTa excels at text completion and SuperGLUE tasks.

**Strengths:**

1. This paper argues against the recent exclusive focus on scaling autoregressive language models, showing instead that masked language models also have significant potential as generative in-context learners. Relatedly, it is demonstrated that these two architectures are better for very different types of tasks. This could inform future efforts in training large-scale systems.
2. The limitations are acknowledged explicitly for each of the proposed techniques, and are all written in a constructive way that suggest clear future directions.
3. The writing and visuals are clear and enjoyable to read.

**Weaknesses:**

1. The text ranking procedure seems unfair to autoregressive models: the MLMs are given access to the right context, except for the two tokens that directly follow the one being predicted. This would give them unfair advantages on certain tasks that require more long-term dependency resolution, or which would benefit from lookahead (e.g., syntactic evaluation tasks); thus, the comparison seems a bit unprincipled in these tasks. Would it be possible to devise a technique that does not give the model any of the right context (e.g., shortening the end of the sequence to just [MASK] [MASK] [MASK] [SEP] at each prediction step, instead of giving the rest of the tokens before [SEP])? This would remove any semantic hints.
2. The proposed inference techniques are model-specific. Thus, future efforts using MLMs will need to take into account the quirks of the MLM objective that a particular model uses when writing pseudo-log-likelihood estimation functions. The authors explicitly acknowledge this limitation.
3. Using OPT as a comparison to DeBERTa in the length generalization experiments seems unfair: this model is informally known to perform poorly on many tasks compared to more recent architecturally similar models like Pythia or Llama. Thus, presenting these results side-by-side could mislead readers into believing that MLMs are unilaterally capable of length generalization while autoregressive models are not. I know the stated point of this paper is to compare models released around the same time, but actually, the paper seems to be about comparing different types of LM objectives at performing generative in-context learning. Thus, I think using some of the models from the cited RULER paper would be an acceptable and more fair comparison---or even just a set of Llama or Alpaca models.
4. It is repeatedly stated that hybrid architectures making use of both autoregressive and masked LM elements is a promising future direction. However, no real evidence is given for this. It is true that MLMs and autoregressive models excel at different things, but why would this mean that combining them would give us the best of both worlds, as opposed to the worst (or simply some mean interpolation between their performances, rather than an overall gain over both)? A few papers are cited as examples, but these seem less like hybrid models and more like entirely distinct ideas (e.g., T5 is cited, but encoder-decoder models like these have different pros and cons even relative to encoder-only or decoder-only models).

**Questions:**

Questions:
* When computing PLL for sequence ranking, would it be possible to simply remove all of the right context and give three MASK tokens followed by SEP at each probability estimation?
* Could you describe some specific ways in which one could combine autoregressive and masked LMs to truly get the best of both worlds? My (admittedly naïve) ideas about how one would do this all involve some mixture of objectives during pre-training. This seems like it would implicitly end up just giving you the advantages of MLMs, since a lot of their behavior comes from the fact that they receive more information (from the right context) than autoregressive systems.

Suggestions/Typos:
* L15-16: inter alig -> inter alia
* When discussing MLM scoring (Salazar et al., 2020), could you also cite the improved version introduced by Kauf & Ivanovna (2023) here: https://aclanthology.org/2023.acl-short.80/

**Limitations:**

The limitations of directly comparing models that perform inference with right context versus without could be more directly discussed. Otherwise, I think the authors have done a good job of discussing limitations.

---

> ### Author Rebuttal · Authors · 2024-08-07
>
> Thank you for the review!
> ____
> > The text ranking procedure seems unfair to autoregressive models: the MLMs are given access to the right context, except for the two tokens that directly follow the one being predicted. This would give them unfair advantages on certain tasks that require more long-term dependency resolution, or which would benefit from lookahead (e.g., syntactic evaluation tasks); thus, the comparison seems a bit unprincipled in these tasks. Would it be possible to devise a technique that does not give the model any of the right context (e.g., shortening the end of the sequence to just [MASK] [MASK] [MASK] [SEP] at each prediction step, instead of giving the rest of the tokens before [SEP])? This would remove any semantic hints.
>
> We completely agree with this point, except for your statement that it is *unfair* and a weakness :) Both models are given the full context, but CLMs are limited to process it left-to-right, which indeed seems (and most likely is) suboptimal. MLMs do not mask-out half of the attention matrix and that can definitely be beneficial for some tasks. See below for your suggested ranking method.
> ____
> > Using OPT as a comparison to DeBERTa in the length generalization experiments seems unfair: this model is informally known to perform poorly on many tasks compared to more recent architecturally similar models like Pythia or Llama. Thus, presenting these results side-by-side could mislead readers into believing that MLMs are unilaterally capable of length generalization while autoregressive models are not.
>
> Thank you for this point, this is not the how we expected these experiments to be interpreted, we will need to frame it more clearly. From our point of view, we decided to include the length generalization experiment because this ability of DeBERTa was absolutely essential to demonstrate in-context learning (in the end, this was the reason why we chose DeBERTa instead of other MLMs). Since the overall theme is comparison to GPT-3, it felt natural to also compare the two models here. But indeed, what is really compared is absolute positional encoding (in GPT-3) and relative positional encoding (in DeBERTa), not CLM vs. MLM, we will make it more clear.
> ____
> > When computing PLL for sequence ranking, would it be possible to simply remove all of the right context and give three MASK tokens followed by SEP at each probability estimation?
>
> Absolutely, we will include this as an ablation study in the Appendix, it is the last row in the following table:
> |Ranking method|ReCoRD (EM)|ReCoRD (F₁)|
> |--------------|-----------|-----------|
> |PLL; 1 mask (original PLL)|80.9|81.6|
> |PLL; 2 masks|86.0|86.8|
> |PLL; 3 masks (our method)|87.1|87.9|
> |PLL; 4 masks|86.9|87.8|
> |Exact log-likelihood |77.2|77.8|
> ____
> > Could you describe some specific ways in which one could combine autoregressive and masked LMs to truly get the best of both worlds? My (admittedly naïve) ideas about how one would do this all involve some mixture of objectives during pre-training. This seems like it would implicitly end up just giving you the advantages of MLMs, since a lot of their behavior comes from the fact that they receive more information (from the right context) than autoregressive systems.
>
> As we hinted in the paper, the MLM training objective can be improved to be better for generation by two modifications: removing the last `[SEP]` token and shifting the outputs by one to the right. Then it should also be easy to do the mixture of objectives, as you suggest, which would make the resulting LMs more flexible during inference, if nothing else.
> ____
> > When discussing MLM scoring (Salazar et al., 2020), could you also cite the improved version introduced by Kauf & Ivanovna (2023) here.
>
> Thank you for this link, we were not familiar with this work, which indeed seems relevant and should be cited.

---

> > ### Comment · Reviewer_7Ges · 2024-08-08
> >
> > Thanks for the response, and the additional experiment.
> >
> > I should better articulate my concern: I think on tasks where the aim is to match human behavior or performance, MLMs cannot be fairly compared to autoregressive models, as they receive more information than autoregressive models or humans would receive when processing information for the first time. Consider a task where one must generate a grammatical completion to a sentence, for example; here, access to the right context gives it an unfair advantage over humans, and over models that process inputs left-to-right. That said, for the classification tasks used in this study—where superhuman performance is desirable—I think you're right. I think the text completion tasks largely fall into the former category, so I think this concern stands there; however, all other tasks fall into the latter category. I think if a discussion of this consideration could be added to the paper, I'll consider this addressed.
> >
> > The new results are interesting! Looks like this does significantly drop performance to well below GPT-3's, but it's still pretty capable. I'll be curious to see how this looks for other tasks. Maybe this is attributable to the train-test mismatch compared to the autoregressive model; if so, your hybrid training idea could potentially get it the rest of the way there. :)
> >
> > I still think this is a good paper that will cause people to rethink the current preeminence of autoregressive models. However, I'm still not entirely convinced that some of these comparisons are fully principled, or when differences between models should be attributed to the architecture/inference techniques as opposed to other differences between systems. I'm therefore keeping my original positive score.

---

> > > ### Author Response · Authors · 2024-08-12
> > >
> > > > Consider a task where one must generate a grammatical completion to a sentence.
> > >
> > > Thank you for the thought-provoking response! Just to clear a potential misunderstanding -- note that when it comes to generation, the model produces the output autoregressively, without access to the right context (as there isn't any); but the left context (prompt + already generated text) is processed bidirectionally (see Figure 2, Text Generation). So it doesn't have access to more information than a CLM, it just doesn't limit itself from processing the available information (left context) bidirectionally.
> > >
> > > > I think if a discussion of this consideration could be added to the paper, I'll consider this addressed.
> > >
> > > Yes, we will try to clarify the generation process in the final version and discuss the difference of bidirectional processing.

---

### Official Review · Reviewer_ayxu · 2024-07-13

**Soundness:** 3
**Presentation:** 3
**Contribution:** 3
**Rating:** 7
**Confidence:** 4

**Summary:**

This paper investigates whether BERT style masked language models can perform in-context learning in multiple LM benchmarks used in the GPT-3 paper, and specifically compares the results of DeBERTa to the GPT-3 model. The paper shows that in multi-choice Q&A, winogrand style quizzes the DeBERTA model can match (sometimes outperforms) GPT-3 model, whereas DeBERTa down performs in machine translation benchmarks. To be able to evaluate these models on generative tasks, the paper also introduces a simple way to use BERT like models in auto-regressive generation, which is itself a significant challenge in the field.

**Strengths:**

- The paper presents extensive results for MLMs for in-context learning which were missing and highly required in the field.
- The paper proposes simple (slightly hacky) procedures for (i) auto-regressive sampling from MLMs, (2) calculating logprobs of a sequence of tokens with MLMs.

**Weaknesses:**

- The comparison of sampling procedures missing. I would like to see what scores you would get if you calculate logprobs by just summing individual pseudo-likelihoods of the tokens without doing +2 additional mask tokens. Similarly, what happens if you use Wang and Cho, 2019 method for text generation?
- The few-shot scaling curves are not complete: only provided 0-shot, 1-shot, and the best few-shot result.

**Questions:**

- In machine translation benchmarks, were there any examples above L>512, and what the reported results did for those examples for GPT-3?
- In Figure 3, you write “[w]e use an open-source replication of GPT-3, OPT, which should perform similarly on this task”. How did you know this? I would delete this claim unless you discuss the evidence. I think it would be unnecessary to discuss it here. So, I suggest removing that claim.
- Figure 4 is a little odd. Why don’t you have an actual scaling curve with the actual number of shots but instead have 0, 1, and few labels?

**Limitations:**

As noted in the paper, The sampling procedure is too slow to use in practice since for each token the model needs to process the entire sequence again due to bi-directional attention.

---

> ### Author Rebuttal · Authors · 2024-08-07
>
> Thank you for your review!
> ____
> > The comparison of sampling procedures missing. I would like to see what scores you would get if you calculate logprobs by just summing individual pseudo-likelihoods of the tokens without doing +2 additional mask tokens. Similarly, what happens if you use Wang and Cho, 2019 method for text generation?
>
> We will include an ablation experiment for the proposed ranking method that also addresses comparison. The main problem of Wang and Cho (2019) was consistency in our preliminary experiments, we will compute more quantitative results in the next few days.
>
> |Ranking method|ReCoRD (EM)|ReCoRD (F₁)|
> |--------------|-----------|-----------|
> |PLL; 1 mask (original PLL)|80.9|81.6|
> |PLL; 2 masks|86.0|86.8|
> |PLL; 3 masks (our method)|87.1|87.9|
> |PLL; 4 masks|86.9|87.8|
> |Exact log-likelihood (used in causal LM) |77.2|77.8|
> ____
> > The few-shot scaling curves are not complete: only provided 0-shot, 1-shot, and the best few-shot result.
>
> We were limited by the official GPT-3 results, which were done in the exact same way (https://arxiv.org/abs/2005.14165, Table H.1). But this is a good point, we will include additional data points for the few-shot experiments in a separate Appendix.
> ____
> > In machine translation benchmarks, were there any examples above L>512, and what the reported results did for those examples for GPT-3?
>
> All machine-translation experiments were done with 16 shots, if we look at German-English translation, for example, the average length of the whole prompt is 1113 with std of 147 (this also demonstrates the need of length generalization beyond 512 tokens). It is impossible to comment on the GPT-3 results, all we know are the final scores and that they used 64 shots to get them.
> ____
> > In Figure 3, you write “[w]e use an open-source replication of GPT-3, OPT, which should perform similarly on this task”. How did you know this? I would delete this claim unless you discuss the evidence. I think it would be unnecessary to discuss it here. So, I suggest removing that claim.
>
> Because OPT uses the same transformer architecture as GPT-3, in particular, it uses absolute positional encoding. This strictly limits any model from generalizing to longer inputs than trained on -- thus we can confidently say that GPT-3 is not able to process longer inputs than 2048, exactly like OPT. This is different from modern LMs, which usually use rotary positional encodings, or from DeBERTa with relative positional encoding. We definitely agree that this should be explained more clearly in the paper.
> ____
> > Figure 4 is a little odd. Why don’t you have an actual scaling curve with the actual number of shots but instead have 0, 1, and few labels?
>
> As commented above, this is because we decided to follow the methodology from the GPT-3 paper, which uses different few-shot settings for different tasks. This makes sense in practice, because the average length of samples differs a lot between datasets (entire documents in reading comprehension vs. sentences in machine translation). We believe it makes sense to keep this Figure as it is, to not lose the ability to compare against GPT-3, but we will add an additional Appendix section with your suggested approach.

---

> > ### Author Response · Authors · 2024-08-10
> >
> > As promised, here is the additional experiment that compares different generation methods measured on German->English translation (BLEU, 1 shot). We were not able to get consistent outputs with the method from Wang and Cho (2019), even after trying different configurations that could be more suitable for DeBERTa; the result below follows the configuration from their official GitHub repository. The outputs usually look like this: `It's,,,,,,,. going to be,,,,. the.. is, or,,,,. the.. is, or,,,,,,. it's,,,,,,,. the.. is,,,,,.`, with many repeated high-frequency tokens.
> >
> > |                                                 | BLEU |
> > |-------------------------------------------------|------|
> > | Autoregressive generation (1 mask)              | 9.2  |
> > | Autoregressive generation (2 masks)             | 21.3 |
> > | Autoregressive generation (3 masks, our method) | 23.7 |
> > | Wang and Cho (2019)                             | 0.4  |

---

> > ### Comment · Reviewer_ayxu · 2024-08-11
> > **thank you**
> >
> > Thank you I read the rebuttal. I will keep my positive score as is.

---

### Official Review · Reviewer_iPtc · 2024-07-14

**Soundness:** 2
**Presentation:** 2
**Contribution:** 2
**Rating:** 3
**Confidence:** 4

**Summary:**

This work proposes a simple modification on masked LMs such that they can be used as a generative way and conduct experiments as generative models. The claim of this work is that, through this modification, masked LMs can be used as in-context learners as GPT-3, adapting to new tasks with further fine-tuning. The experiments demonstrate the performance improvement with DeBERTa and a few categories of NLP tasks.

**Strengths:**

Overall, this work demonstrate the possibility of using masked LMs in a generative way. Although the modification is simple, the experiment results on some benchmark evaluation tasks are impressive.

In addition, the method description with limitations show the full picture of the proposed methods, which gives readers a comprehensive understanding about this work.

**Weaknesses:**

Despite of the impressive results, the work and the experiments are still in the preliminary stage, where more work can be done or needed to be done. For example

- The proposed modification may be too simple, and more technical modification is needed to improve the efficiency. For example, in section 2.1, the opportunity of providing a more technically solid work is simplified as “We believe that this limitation can be fixed, …” Intuitively, I agree, although I do expect more technical novelty from this part.
- Similar concern also applies to section 2.2, where it says “We improve on this behavior by interpolating between …”. Again, more technical discussion and novelty on this part will be deeply appreciated.

Without appropriate technical depth, I am wondering what the technical novelty of this work could be.

In addition, the experiment design is less convincing. The title and most of the discussion are about masked LMs, while the experiments are only about DeBERTa.

There are also some writing issues about this paper, for example,

- Although this paper is about in-context learning, the major modification is about turning masked LMs to generative models. So, it’s not clear to me how this can be particularly tied to in-context learning
- What the notation means in equations 1 and 2?
- I think there may be some typos in equation 2

About experiments

- It would be great to list the inference costs of DeBERTa and GPT-3
- What are the values of $k$ for few-shot learning?

**Questions:**

Please refer to the previous section.

**Limitations:**

The paper has listed the limitations of the proposed methods, for example, how to improve the proposed algorithms. However, given the lack of technical depth, some of the listed limitations should be addressed in this work, instead of future work.

---

> ### Author Rebuttal · Authors · 2024-08-07
>
> Thank you for the time and effort spent on this review. While we may not completely agree with all your points (see below), they will be very helpful in making the paper more clear in the next version.
> ____
> > The proposed modification may be too simple, and more technical modification is needed to improve the efficiency. For example, in section 2.1, the opportunity of providing a more technically solid work is simplified as “We believe that this limitation can be fixed, …” Intuitively, I agree, although I do expect more technical novelty from this part.
>
> We believe that this is out of scope of this paper as its main focus is to demonstrate in-context learning of MLMs. Doing a good job at optimizing the efficiency of the generative method would most likely involve some changes to the MLM training objective, which is definitely an interesting topic, but it should be addressed in a separate paper, it is not appropriate for a footnote in this paper.
> ____
> > Similar concern also applies to section 2.2, where it says “We improve on this behavior by interpolating between …”. Again, more technical discussion and novelty on this part will be deeply appreciated.
>
> We will expand on this in an extra section in the Appendix that will describe the ablation experiments for our proposed method. In a nutshell, since the problem of the original PLL is in estimating likelihoods of long multi-loken expressions, we ablate this "interpolation" on the ReCoRD dataset, which involves answers with long named entities.
>
> |Ranking method|ReCoRD (EM)|ReCoRD (F₁)|
> |--------------|-----------|-----------|
> |PLL; 1 mask (original PLL)|80.9|81.6|
> |PLL; 2 masks|86.0|86.8|
> |PLL; 3 masks (our method)|87.1|87.9|
> |PLL; 4 masks|86.9|87.8|
> |Exact log-likelihood (used in causal LM) |77.2|77.8|
> ____
> > In addition, the experiment design is less convincing. The title and most of the discussion are about masked LMs, while the experiments are only about DeBERTa.
>
> Our main claim is that MLMs can function as in-context learners, for this, it is enough to show that one such model exhibits these abilities. There are technical reasons for choosing DeBERTa over other models, we describe them in Related Work as well as in Section 2. In short, we need a model that is capable of processing long inputs (> 512 tokens), that is large (>1B parameters) and that is primarily trained on English. As far as we know, DeBERTa is the only model that satisfies this.
> ____
> > Although this paper is about in-context learning, the major modification is about turning masked LMs to generative models. So, it’s not clear to me how this can be particularly tied to in-context learning.
>
> All 1-shot and few-shot experiments clearly demonstrate in-context learning. Being able to generate text is necessary to show in-context learning in tasks such as machine translation. We don't think that we fully understand your point, could you please elaborate on this comment if we didn't address it?
> ____
> > What the notation means in equations 1 and 2? I think there may be some typos in equation 2.
>
> There is indeed a small typo, the last $w_{i−1}$ should be $w_{i+1}$, thanks! In terms of variables and notation used, $w_0 \oplus w_1 \dots w_k$ is a completion of a prompt $c$, then Equation 1 is just a simple chain rule while Equation 2 describes approximation of the previous equation with MLM; we will introduce this notation better in the future version.
> ____
> > It would be great to list the inference costs of DeBERTa and GPT-3.
>
> We didn't include it because we were not sure if it would be beneficial for the reader or not, we rely on the HF implementation of DeBERTa, which is (in our opinion) far from optimized; instead we include more general limitations. However, we also see the benefit of providing concrete numbers. When measuring the cost of generating 256 tokens from a 256-long prompt; OPT (~GPT-3) with cache: 3.8s/it, OPT without cache: 12.4s/it, DeBERTa (without cache): 20.2s/it.
> ____
> > What are the values of $k$ for few-shot learning?
>
> We completely agree that this should be described in the paper, we will put this information together with all results into a separate table in Appendix, in a similar fashion to the original GPT-3 paper. For completeness, this is the table with all results and few-shot values:
>
> | Task | Split | Metric | n shots | 0-shot (1.4B) | 1-shot (0.1B) | 1-shot (0.4B) | 1-shot (0.9B) | 1-shot (1.4B) | n-shot (1.4B) |
> |------|-------|--------|---------|---------------|---------------|---------------|---------------|---------------|---------------|
> |BoolQ|dev|acc.|4|80.8|55.7|60.5|78.4|82.1|82.1|
> |CB|dev|acc.|4|66.1|39.6|57.5|68.6|76.1|75.0|
> |CB|dev|F₁|4|46.1|23.8|39.8|47.1|57.0|57.6|
> |COPA|dev|acc.|64|78.9|67.0|78.0|80.6|84.2|90.4|
> |MultiRC|dev|EM acc.|4|6.6|2.4|7.0|11.1|15.6|16.9|
> |MultiRC|dev|F₁ₐ|4|61.6|57.2|57.4|57.0|67.9|69.2|
> |ReCoRD|dev|EM acc.|4|87.1|62.3|73.6|86.8|87.4|87.4|
> |ReCoRD|dev|F₁|4|87.9|63.0|74.3|87.5|88.1|88.2|
> |RTE|dev|acc.|8|64.3|50|53.1|64.5|65.0|62.2|
> |WiC|dev|acc.|16|55.2|49.6|49.6|49.6|50.3|50.2|
> |WSC|dev|acc.|16|71.2|62.3|65.0|67.3|69.6|75.0|
> |HellaSwag|dev|acc.|16|62.0|36.9|51.3|58.7|62.4|62.5|
> |StoryCloze|test|acc.|32|83.6|69.5|77.0|82.4|84.6|84.8|
> |Winograd|test|acc.|32|74.0|59.3|68,1|76.2|80.7|85.6|
> |Winogrande|dev|acc.|32|61.0|49.8|54.8|60.6|63.6|68.8|
> |DE--EN|test|BLEU|16|2.4|0.2|4.6|20.0|23.7|25.1|
> |EN--DE|test|BLEU|16|1.6|0.2|0.4|3.4|5.4|6.6|
> |FR--EN|test|BLEU|16|1.7|0.2|8.7|21.9|23.5|24.5|
> |EN--FR|test|BLEU|16|0.3|0.0|0.3|4.6|9.7|10.8|
> |RO--EN|test|BLEU|16|1.7|0.2|4.3|16.0|17.7|18.9|
> |EN--RO|test|BLEU|16|0.1|0.0|0.2|1.2|2.5|4.1|
> |Natural Questions|test|EM acc.|16|0.8|0.1|0.6|2.1|2.6|4.4|
> |TriviaQA (wiki)|dev|EM acc.|16|6.9|0.9|3.8|13.6|14.3|17.9|
> |Web Questions|test|EM acc.|32|1.5|0.3|1.0|4.5|5.1|9.9|
> |PIQA|dev|acc.|32|72.9|62.4|69.6|71.6|73.0|74.5|
> |ARC (challenge)|test|acc.|32|36.5|25.3|33.2|35.9|37.1|39.6|
> |ARC (easy)|test|acc.|32|55.1|39.6|46.3|53.3|55.1|57.7|
> |OpenBookQA|test|acc.|96|45.8|35.0|41.8|42.8|46.4|50.4|

---

> > ### Comment · Reviewer_iPtc · 2024-08-12
> > **Thanks for the additional information, still concerned about technical novelty**
> >
> > As I mentioned in the title, I appreciate the additional clarification and results, some of which directly answer my questions. However, I am still concerned about the technical novelty of this work. Since efficiency is the core of generative models, asking the efficiency question when we test the MLM capacity on ICL is reasonable. Therefore, I don't think the "out of scope" argument makes sense.

---

### Author Response · Authors · 2024-08-14
**Rebuttal revisions**

We would like to thank the reviewers for their time and valuable comments. Based on the feedback, we have made several minor revisions to our paper, which are summarized here. We believe the paper greatly benefited from incorporating their feedback.

* Reviewers *iPtc* and *7Ges* pointed out that the paper is missing evaluation of different sequence-scoring methods. We included an ablation study in the Appendix that quantitatively backs up our proposed method from Section 2.2.
* Similarly, reviewer *ayxu* requested a comparison of different generative methods, we conducted another ablation study that is now included in the Appendix. We think that these two extra evaluations are a great addition to the paper.
* We fixed a typo in Equation 2, as pointed out by reviewer *iPtc*.
* We included a table with all results, together with the values of *k* (number of shots, as requested by reviewer *iPtc*).
* Following the feedback from *ayxu*, a more fine-grained evaluation of few-shot abilities on SuperGLUE is put into the Appendix (similar to Figure 3.8 from [Brown et. al (2020)](https://arxiv.org/abs/2005.14165)).
* We cite [Kauf & Ivanovna (2023)](https://aclanthology.org/2023.acl-short.80/), after reading this paper suggested by reviewer *7Ges*. We also plan to evaluate this method and include it in the ablation study mentioned in the first point.
* We greatly value that the reviews highlighted some potential misunderstandings, we tried to clarify the wording in the key places of the paper.
* We added a section with examples of text generation into the Appendix, to better show the generative ability of masked language models. We again follow the [Brown et. al (2020)](https://arxiv.org/abs/2005.14165) for these qualitative tests and show samples from "learning and using novel words" test and "correcting English grammar" test.

---

### Decision · Program_Chairs · 2024-09-25

**Decision:**

Accept (poster)

**Comment:**

This paper seeks to demonstrate that MLMs can be capable in-context learnings, comparable with their autoregressive counterparts. The paper proposes a simple (if somewhat inefficient) technique for sampling autoregressively from DeBERTa. Experiments with DeBERTa and a matched autoregressive LM (GPT-3) are conducted on a suite of in-context learning evaluations. Surprisingly, the MLM (DeBERTa) performs well on in-context learning tasks and in some cases outperforms GPT-3. The majority of reviewers found the results of this paper both surprising and valuable, indicating that the demonstration of ICL ability in MLMs (showing advantages over autoregressive models on some tasks) might point to new training objectives and learning strategies for LLMs more generally. Some concerns were raised about the focus on DeBERTa (rather than other MLMs) and whether DeBERTa had unfair access to right-hand context in experiments -- these concerns were all adequately addressed in rebuttal.